# Identifying modifiable risk factors for mortality in children aged 1–59 months admitted with WHO-defined severe pneumonia: a single-centre observational cohort study from rural Malawi

Michelle Eckerle,[1] Tisungane Mvalo,[2,3] Andrew G Smith,[4] Davie Kondowe,[2] Don Makonokaya,[2] Dhananjay Vaidya,[5] Mina C Hosseinipour,[2,6] Eric D McCollum [7,8]

For numbered affiliations see end of article.

**Correspondence to**
Dr Eric D McCollum; emccoll3@jhmi.edu

## ABSTRACT

**Objective** Although HIV infection, severe malnutrition and hypoxaemia are associated with high mortality in children with WHO-defined severe pneumonia in sub-Saharan Africa, many do not have these conditions and yet mortality remains elevated compared with high-resource settings. Further stratifying mortality risk for children without these conditions could permit more strategic resource utilisation and improved outcomes. We therefore evaluated associations between mortality and clinical characteristics not currently recognised by the WHO as high risk among children in Malawi with severe pneumonia but without HIV (including exposure), severe malnutrition and hypoxaemia.

**Methods** Between May 2016 and March 2018, we conducted a prospective observational study alongside a randomised controlled trial (CPAP IMPACT) at Salima District Hospital in Malawi. Children aged 1–59 months hospitalised with WHO-defined severe pneumonia without severe malnutrition, HIV and hypoxaemia were enrolled. Study staff assessed children at admission and ascertained hospital outcomes. We compared group characteristics using Student's t-test, rank-sum test, $\chi^2$ test or Fisher's exact test as appropriate.

**Results** Among 884 participants, grunting (10/112 (8.9%) vs 11/771 (1.4%)), stridor (2/14 (14.2%) vs 19/870 (2.1%)), haemoglobin <50 g/L (3/27 (11.1%) vs 18/857 (2.1%)) and malaria (11/204 (5.3%) vs 10/673 (1.4%)) were associated with mortality compared with children without these characteristics. Children who survived had a 22 g/L higher mean haemoglobin and 0.7 cm higher mean mid-upper arm circumference (MUAC) than those who died.

**Conclusion** In this single-centre study, our analysis identifies potentially modifiable risk factors for mortality among hospitalised Malawian children with severe pneumonia: specific signs of respiratory distress (grunting, stridor), haemoglobin <50 g/L and malaria infection. Significant differences in mean haemoglobin and MUAC were observed between those who survived and those who died. These factors could further stratify mortality

### What is known about the subject?

► Lower respiratory infections like pneumonia kill an estimated 800 000 children annually, making it the number one infectious cause of paediatric deaths globally.
► Among children with pneumonia, important drivers of mortality in low and middle-income countries (LMICs) are severe malnutrition, HIV infection or exposure and hypoxaemia.
► Children in LMICs with severe pneumonia lacking HIV, malnutrition and hypoxaemia may represent a large proportion of pneumonia burden and mortality but data are limited.

### What this study adds?

► About 20% of pneumonia deaths at one rural district hospital in Malawi over 2 years were attributable to children lacking HIV, severe malnutrition and hypoxaemia.
► Among Malawian children with severe pneumonia without HIV, severe malnutrition and hypoxaemia, respiratory distress (stridor, grunting), haemoglobin <50 g/L and malaria predicted mortality.
► Significant differences in mean haemoglobin and mid-upper arm circumference were present in severe pneumonia cases lacking recognised high-risk conditions who survived or died.

risk among hospitalised Malawian children with severe pneumonia lacking recognised high-risk conditions.

## INTRODUCTION

Pneumonia causes approximately 800 000 deaths outside of the neonatal period among under 5-year-olds worldwide and this burden

is disproportionately borne in low and middle-income countries (LMICs).[1] Severe malnutrition, HIV infection or exposure and hypoxaemia (peripheral oxyhaemoglobin saturation ($SpO_2$) <90%) place children with pneumonia at higher risk of death.[1–4] While children with severe pneumonia but without these conditions may have relatively lower mortality, they still may comprise a substantial overall burden of pneumonia morbidity and mortality in LMICs.[1] For example, in Malawi, approximately half of child pneumonia deaths were attributable to HIV and almost 90% of hospitalised pneumonia were among children without hypoxaemia.[4 5] In a multi-country severe pneumonia study, 61% of deaths occurred among children without severe malnutrition.[6] Identifying clinical characteristics that predict mortality, can be objectively measured and may be modifiable among paediatric pneumonia cases lacking severe malnutrition, HIV (including exposure) and hypoxaemia could represent an effective strategy for lowering pneumonia deaths in LMICs.

The WHO recommends a syndromic approach to child pneumonia diagnosis and management in LMICs to improve implementation feasibility.[7] Recommended treatment for WHO-defined severe pneumonia includes hospitalisation, parenteral antibiotics and oxygen.[7] Understanding which children with WHO-defined severe pneumonia in LMICs are at high risk of death is important as it can impact resource allocation in austere settings where advanced care and other resources are limited.[8]

In conjunction with a randomised controlled trial called CPAP IMPACT, which studied children with WHO-defined severe pneumonia and at least one high-risk condition (severe malnutrition, HIV infection or exposure and/or hypoxaemia),[9 10] we conducted a parallel prospective observational study of children with WHO-defined severe pneumonia but without any of these three high-risk conditions. Our main objectives were to contextualise observational study participants through their clinical characteristics and outcomes, and then to identify clinical predictors of mortality with a priority towards factors that may be objectively measured, potentially modifiable and not included in WHO pneumonia guidelines.

## METHODS

We conducted a prespecified substudy in conjunction with a randomised controlled trial (CPAP IMPACT).[9 10] CPAP IMPACT compared bubble continuous positive airway pressure to low flow oxygen for children with WHO-defined severe pneumonia and at least one high-risk condition. From May 2016 through March 2018, children with WHO-defined severe pneumonia and ineligible for CPAP IMPACT were invited to participate in this ancillary prospective observational study. The study took place at Salima District Hospital in Salima, Malawi, a secondary hospital lacking intensive care resources. Enrolled participants provided written consent.

During the study period, children admitted to Salima District Hospital were screened for eligibility at all times. Children were eligible if aged 1–59 months with WHO-defined severe pneumonia and without any of three high-risk conditions: (1) HIV infection (<12 months old with a positive HIV DNA PCR or ≥12 months old with HIV antibodies) or exposure (<24 months old born to an HIV-infected mother and not meeting HIV infection criteria), (2) severe malnutrition (weight-for-age z-score < −3 SD, mid-upper arm circumference (MUAC) <11.5 cm and/or bilateral pedal oedema), or (3) $SpO_2$ <90%.

Per WHO guidelines, children could meet severe pneumonia criteria by having cough and/or difficult breathing plus signs of pneumonia with at least one general danger sign or signs of respiratory distress with or without a general danger sign.[11] Respiratory distress included head nodding, nasal flaring, grunting, stridor when calm, tracheal tugging, severe chest indrawing, very fast breathing for age (≥80 breaths/min if aged 30–59 days, >70 breaths/min if aged 2–11 months and ≥60 breaths/min if aged 12–59 months) or apnoea. Signs of pneumonia included fast breathing (60–79 breaths/min if aged 30–59 days, 50–69 breaths/min if aged 2–11 months and 40–59 breaths/min if aged 12–59 months) and/or chest indrawing. Inability to feed, vomiting everything, lethargic or unconscious (Blantyre Coma Score ≤4)[12] and convulsions comprised general danger signs.

Consistent with national guidelines, malaria infection was confirmed on all participants at enrolment using rapid antigen testing (SD Bioline (HRP2/pLDH, sensitivity 97.4%–99.7%, specificity 99.3%–99.7%)) as routine microscopy was limited.[13] Anaemia was also assessed by point-of-care testing (HemoCue 301+). A Rad-5 pulse oximeter (Masimo, Irvine, California, USA) measured the $SpO_2$ on children breathing in room air. Study staff recommended initial evaluations and treatments at enrolment and thereafter care was led by government hospital staff, including oxygen, antibiotics and fluids. According to local guidelines, all patients received benzylpenicillin and gentamicin and could be switched to ceftriaxone for treatment failure.[14] Similarly, per local guidelines, severe malaria was treated with artesunate and uncomplicated malaria by lumefantrine-artemether.[13] Children were followed passively by study staff until hospital outcome. This study's primary outcome was in-hospital mortality.

### Statistical analysis

Bivariable analysis comparing clinical variables with in-hospital mortality was performed as follows: the Student's t-test was used to assess differences between the means of normally distributed continuous variables (tabulated as means and SDs), the non-parametric Mann-Whitney U test evaluated differences in the rank-sums of continuous variables (tabulated as medians and IQR presented as 25th and 75th percentiles, $\chi^2$ and Fisher's exact tests (if the large sample assumption could not be

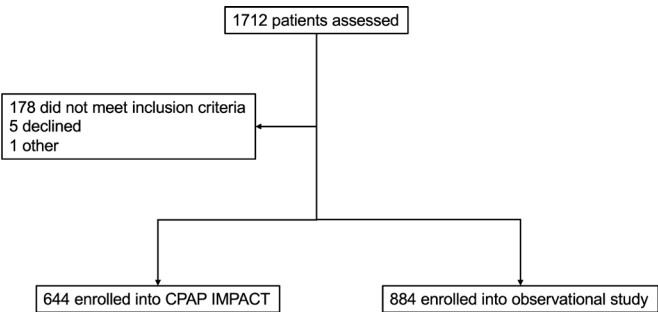

**Figure 1** Study profile.

justified) analysed categorical clinical variables (tabulated as numbers and percentages)). To understand whether an association existed between mortality and the number of significant risk factors per patient, logistic regression was performed. Although we planned to fit multivariable regression models for mortality we determined it was not advisable due to the number of mortality events experienced. All analyses were conducted using Stata (V.15.1; StataCorp, College Station, Texas).

### Patient and public involvement
Patients and the public in Malawi were sensitised to the development, design, recruitment and conduct via local meetings before, during and after the research.

### RESULTS
From May 2016, enrolment began for the observational study and concluded at the same time as the parent CPAP IMPACT trial in March 2018 (figure 1). Among 1062 children who did not meet inclusion criteria for CPAP IMPACT, 884 met criteria for the observational study and agreed to participate.

We sought to identify whether any participant characteristics—with an emphasis on objective and potentially modifiable factors not included in the WHO pneumonia guidelines[7]—had associations with hospital mortality (table 1). Children with respiratory distress signs like grunting (10/112 (8.9%) vs 11/771 (1.4%), p<0.01) and stridor when calm (2/14 (14.2%) vs 19/870 (2.1%), p=0.04) had higher mortality rates than children without these signs. Anaemia and malaria were also associated with mortality. Specifically, children who died, compared with survivors, had a 22 g/L lower mean haemoglobin (78 g/L (SD 30) vs 100 g/L (SD 20), p<0.01). Children with a haemoglobin <50 g/L also had higher mortality than those with a 50 g/L or higher measurement (11.1% vs 2.1%, p≤0.02). Children positive for malaria had a mortality of 5.4% while children testing negative had a mortality rate of 1.4% (p<0.01). Children with malaria, compared with those testing negative, had a 19 g/L lower mean haemoglobin level (85 g/L vs 104 g/L, p<0.01). In a sensitivity analysis excluding seven children without a malaria test result, the overall findings were consistent (online supplemental table 1).

Participants who died had a 1.7 cm lower mean MUAC than those who did not die (p<0.01). Notably, there was no difference between those who survived and those who did not based on the admission $SpO_2$ distribution (p=0.49), and although the point estimates for mortality were higher among children with $SpO_2$ of 90%–92%, compared with 93%–100%, these did not reach statistical significance (3.7% vs 2.2%, p=0.42).

Of the identified risk factors for mortality, we considered dichotomous factors (stridor when calm, grunting, malaria status, haemoglobin<50 g/L) and used logistic regression to determine whether the odds of mortality increased with increasing numbers of risk factors per patient. We found that compared with children with only one risk factor, children with two, three or four risk factors had an increased OR for death that increased with an increasing number of risk factors (table 2).

In online supplemental table 2, we compared observational participants to children enrolled in CPAP IMPACT to provide context for both groups as well as the spectrum of pneumonia at Salima District Hospital during the study period. The mortality of observational children was 11.3 absolute percentage points lower than those in CPAP IMPACT (2.4% vs 13.7%, p<0.01). Altogether, when accounting for trial participants and the observational group, the hospital mortality of severe pneumonia was 7.1% (109/1528).

### DISCUSSION
We identified potentially modifiable, objectively measurable risk factors for mortality among an observational cohort of Malawian children with WHO-defined severe pneumonia and without severe malnutrition, HIV or hypoxaemia. We found those children who died were more likely to present with respiratory distress, anaemia, malaria and a lower MUAC. Our results also show that during the study period severe pneumonia was generally as frequent among Malawian children without high-risk conditions as they were with them, and that about 20% of pneumonia deaths at Salima District Hospital during this period were attributable to lower risk children. Assuming these data are representative then potentially high numbers of pneumonia deaths among sub-Saharan African children without severe malnutrition, HIV and hypoxaemia may occur each year.[15] Altogether, if confirmed, our results imply such children could be important in the broader pneumonia epidemiological context and may benefit from more systematic scrutiny of these additional, potentially modifiable domains of risk, (1) respiratory distress, (2) anaemia, (3) malaria, and (4) non-severe malnutrition, through more rigorous screening and testing.

Currently, only MUAC is routinely recommended by the WHO Integrated Management of Childhood Illnesses (IMCI) guidelines for use at frontline health facilities and hospital outpatient clinics serving as an entry point for most patients.[7] Instead, guidelines

**Table 1** Association between hospitalisation characteristics and outcome among children aged 1–59 months with WHO-defined severe pneumonia without HIV, severe malnutrition and/or hypoxaemia in an observational study

| Characteristic | | Survived n=863 | Died n=21 | P value |
|---|---|---|---|---|
| **Demographics** | | | | |
| Age (months), median (IQR)* | | 9.2 (4.7, 17.3) | 8.1 (6.5, 12.0) | 0.74 |
| Sex, n (%)† | Females | 353 (97.5) | 9 (2.5) | 1.00 |
| | Males | 510 (97.7) | 12 (2.3) | |
| **Clinical features** | | | | |
| Respiratory rate (breaths/min), mean (SD)‡ | | 59.6 (12.9) | 60.3 (13.7) | 0.82 |
| Fast breathing for age, n (%)† | Yes | 720 (97.6) | 18 (2.4) | 1.00 |
| | No | 143 (97.9) | 3 (2.1) | |
| Pulse rate (beats/min), mean (SD)‡ | | 162.2 (20.4) | 170.0 (30.4) | 0.09 |
| Pulse rate >180 beats/min, n (%)† | Yes | 113 (95.0) | 6 (5.0) | 0.22 |
| | No | 400 (97.6) | 10 (2.4) | |
| Pulse rate >160 beats/min, n (%)† | Yes | 165 (97.6) | 4 (2.4) | 0.20 |
| | No | 184 (99.5) | 1 (0.5) | |
| Severe chest indrawing, n (%)† | Yes | 760 (97.8) | 17 (2.2) | 0.31 |
| | No | 103 (96.3) | 4 (3.7) | |
| Head nodding, n (%)† | Yes | 180 (95.7) | 8 (4.3) | 0.10 |
| | No | 683 (98.1) | 13 (1.9) | |
| Grunting, n (%)†§ | Yes | 102 (91.1) | 10 (8.9) | <0.01 |
| | No | 760 (98.6) | 11 (1.4) | |
| Stridor when calm, n (%)† | Yes | 12 (85.7) | 2 (14.3) | 0.04 |
| | No | 851 (97.8) | 19 (2.2) | |
| Apnoea, n (%)† | Yes | 6 (85.7) | 1 (14.3) | 0.16 |
| | No | 857 (97.7) | 20 (2.3) | |
| Nasal flaring, n (%)† | Yes | 507 (97.6) | 12 (2.3) | 1.00 |
| | No | 356 (97.5) | 9 (2.5) | |
| **Measurements and laboratory results** | | | | |
| MUAC (cm), mean (SD)‡ | | 14.0 (1.2) | 13.3 (1.0) | <0.01 |
| MUAC 11.5–13.5 cm, n (%)† | Yes | 255 (96.2) | 10 (3.8) | 0.09 |
| | No | 608 (98.2) | 11 (1.8) | |
| Hb (g/L), mean (SD)‡ | | 10.0 (2.0) | 7.8 (3.0) | <0.01 |
| Hb <5 g/L, n (%)† | Yes | 24 (88.9) | 3 (11.1) | 0.02 |
| | No | 839 (99.6) | 18 (2.1) | |
| Malaria positive, n (%)¶ | Yes | 193 (94.6) | 11 (5.4) | <0.01 |
| | No | 663 (98.5) | 10 (1.5) | |
| SpO$_2$ (%), median (IQR)* | | 96 (95, 98) | 96 (94, 98) | 0.49 |
| SpO$_2$† | 93%–100% | 787 (97.8%) | 18 (2.2%) | 0.42 |
| | 90%–92% | 76 (96.2%) | 3 (3.7%) | |

*Wilcoxon rank-sum.
†Fisher's exact test.
‡Two-sample t-test.
§One participant missing data.
¶Seven participants were missing test results.
Hb, haemoglobin; MUAC, mid-upper arm circumference.

recommend using caregiver report to identify difficult breathing, checking for anaemia by examining for pallor and testing for malaria if fever is present (in endemic areas).[7] While severe malnutrition, HIV and hypoxaemia confer a higher mortality risk, there is value in identifying specific features that convey a higher probability of death among lower risk but severely ill patients. It is possible these children, if appropriately identified, may be more treatment responsive and more likely to survive; the ability to target care would be beneficial. For example, such children may be more successfully managed if cohorted into hospital units with higher nurse to patient ratios that allow closer monitoring and more meticulous care. Results from CPAP IMPACT suggest meaningful reductions in mortality among higher risk children with severe pneumonia may be

**Table 2** Association between mortality and number of risk factors (stridor when calm, grunting, haemoglobin <50 g/L, malaria test positive)

| Risk factors (n) | OR (95% CI) | P value |
| --- | --- | --- |
| 1 | Referent | – |
| 2 | 4.5 (1.4 to 14.1) | <0.01 |
| 3 | 14.7 (4.3 to 50.1) | <0.01 |
| 4 | 26.2 (4.4 to 153.7) | <0.01 |

challenging to achieve with available approaches in LMICs.

Previous studies reported similar findings to this study, though analyses aggregate children with and without high-risk conditions or severe and non-severe pneumonia. A large meta-analysis of children from LMICs found that young age (<2 months), *Pneumocystis jirovecii* infection, chronic underlying diseases, HIV/AIDS and severe malnutrition all increased the likelihood of death.[2] Within sub-Saharan Africa, a Kenyan study of paediatric pneumonia hospitalisations sought to establish mortality rates and risk factors among children with pneumonia.[16] Among those with non-severe disease, pallor and a weight-for-age z-score <3 SD were associated with death, but having respirations >70 breaths/min, moderate malnutrition, moderate dehydration or admission in a malaria-endemic region also increased mortality risk. Another Kenyan study of hospitalised children found overall mortality was 5%, and severe anaemia, jaundice, chest indrawing, deep breathing, neurological status and axillary temperature (>39°C or <36°C) were predictive of immediate death.[17]

In our observational cohort, children who died were more likely to have a lower haemoglobin and were also more likely to have malaria than those who survived. As expected, a significant difference in mean haemoglobin existed between children with a positive versus negative malaria test. Given the syndromic approach to disease used by healthcare workers in LMICs, an overlap between primary respiratory illness and malaria has been previously reported. In a prospective cohort of children in Nigerian hospitals, 31% of those with pneumonia also had malaria,[18] while in Mozambique 19% of children with severe pneumonia had malaria.[19] In Uganda, a study of children attending 14 health centres found 30% had both pneumonia and malaria.[20] A large Kenyan cohort of hospitalised hypoxaemic children found a final diagnosis of malaria more frequently than lower respiratory tract infection (35% vs 32%).[21]

Unfortunately, clinical pallor has been shown as unreliable for detecting anaemia. A study of hospitalised Tanzanian children found 8% sensitivity and 98% specificity for pallor identifying severe anaemia.[22] Among children in Uganda and Bangladesh, severe palmar and conjunctival pallor had sensitivities between 10% and 50% and a specificity of 99% for severe anaemia.[23] Altogether, given the coincidence of these conditions, the mortality risk associated with severe malaria and anaemia, and the limited performance of pallor, routine haemoglobin and malaria testing among children with difficult breathing merits further consideration.

Unlike this study, prior inpatient paediatric studies from Malawi found associations between moderately low $SpO_2$ ranges and increased odds of mortality among children with pneumonia (between 50% and 330% higher).[24 25] The paediatric populations in these studies differed as they included severe malnutrition and/or HIV cases. Although in our study we also observed higher point estimates for mortality among children with $SpO_2$ of 90%–92%, compared with 93%–100%, this association did not reach statistical significance. We expect that with a larger sample size, a similar association between $SpO_2$ and mortality would have been observed in this study.

There are several limitations. First, malaria detection was determined by rapid antigen testing rather than peripheral blood microscopy. While children in endemic settings may have a positive rapid test due to persistent antigenaemia rather than an acute infection,[26] our use of rapid testing was consistent with locally recommended practice in Malawi, a setting with limited microscopy capacity.[13] Second, the cohort experienced 21 deaths, and this restricted our ability to fit multivariable models for mortality as planned. Third, although our prospective design and population of rural Malawian children at a district hospital should be considered strengths, this is a single-centre study. Increasing the certainty of our findings remains important and a larger, multicentre study is needed.

In sum, we identified multiple risk factors for death (specific signs of respiratory distress (stridor, grunting), haemoglobin <50 g/L and malaria) among Malawian children with severe pneumonia but without severe malnutrition, HIV infection or exposure and hypoxaemia. Children who died had a lower MUAC than those who survived, but statistically significant thresholds were not apparent. Based on these findings, systematic screening of children with severe pneumonia for additional signs of respiratory distress, checking of haemoglobin levels, testing for malaria (when endemic) and measuring MUAC, followed by closer monitoring and standard treatment for any identified abnormalities, seem feasible and warrant further consideration. Intervention studies examining the modifiability of these risk factors in similar populations are needed as is larger, confirmatory observational research in other LMIC settings.

**Author affiliations**
[1]Division of Emergency Medicine, Department of Pediatrics, Cincinnati Children's Hospital Medical Center, Cincinnati, Ohio, USA
[2]University of North Carolina Project Malawi, Lilongwe, Central Region, Malawi
[3]Department of Pediatrics, School of Medicine, University of North Carolina at Chapel Hill, Chapel Hill, North Carolina, USA
[4]Department of Pediatrics, University of Utah School of Medicine, Salt Lake City, Utah, USA
[5]Department of Pediatrics, BEAD Core, Johns Hopkins University, Baltimore, Maryland, USA

[6]Division of Infectious Disease, Department of Medicine, School of Medicine, University of North Carolina at Chapel Hill, Chapel Hill, North Carolina, USA
[7]Global Program in Pediatric Respiratory Sciences, Eudowood Division of Pediatric Respiratory Sciences, Department of Pediatrics, School of Medicine, Johns Hopkins University, Baltimore, Maryland, USA
[8]Department of International Health, Bloomberg School of Public Health, Johns Hopkins University, Baltimore, Maryland, USA

**Acknowledgements** We thank the children and their caregivers who participated, the Salima District Hospital and the Malawi Ministry of Health for their support of this research and the entire CPAP IMPACT study team for their dedication and care of the children in this study. We also acknowledge the important contributions of Dr Veena Billioux to this work.

**Contributors** EDM acquired the funding. ME and EDM were responsible for conceptualisation and design. EDM curated the data. DK and DM collected the data. DV, ME and EDM were responsible for data analysis. ME, EDM, TM, AGS and MH were responsible for data interpretation. ME and EDM wrote the original draft. ME, EDM, TM, AGS, DK, DM, DV and MH were responsible for writing, review and editing. EDM is the guarantor.

**Funding** The Bill & Melinda Gates Foundation (OPP1123419), the International AIDS Society (141022) and the Health Empowering Humanity provided funding for this work.

**Competing interests** None declared.

**Patient and public involvement** Patients and/or the public were involved in the design, or conduct, or reporting, or dissemination plans of this research. Refer to the Methods section for further details.

**Patient consent for publication** Not required.

**Ethics approval** This study involves human participants and was approved by the Malawi National Health Sciences Research Committee (1325) and the Johns Hopkins Medicine Institutional Review Board (IRB00055734). Participants gave informed consent to participate in the study before taking part.

**Provenance and peer review** Not commissioned; externally peer reviewed.

**Data availability statement** Data are available upon reasonable request.

**ORCID iD**
Eric D McCollum http://orcid.org/0000-0002-1872-5566

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
