## [Reviewer comments · BMJ Paediatrics Open]

ARTICLE DETAILS

TITLE (PROVISIONAL)	Identifying modifiable risk factors for mortality in children aged 1-59 months admitted with WHO-defined severe pneumonia: a single centre observational cohort study from rural Malawi
AUTHORS	Eckerle, Michelle Mvalo, Tisungane Smith, Andrew Kondowe, Davie Makonokaya, Don Vaidya, Dhananjay Hosseinipour, Mina McCollum, Eric D.

VERSION 1 – REVIEW

REVIEWER	Reviewer name: Dr. Peter Flom Institution and Country: Peter Flom Consulting, United States Competing interests: None
REVIEW RETURNED	28-Oct-2021

GENERAL COMMENTS	I confine my remarks to statistical aspects of this paper. The general approach is fine, but I do have some issues to resolve before i can recommend publication. Line 47 "Elevated" compared to what population? Rich countries? Earlier years in Malawi? Other African countries? Line 62-64 Why are you comparing these two groups? I don't see a reason, in which case, these lines can be dropped (the trial participants were selected to have problems). (Similarly for some sections of text. Why compare these groups?) (Not statistics but this would also let the authors drop the word "observational" in many places, leading to a smoother flow of text). Line 98 When you say "leading cause" please give a rate (incidence or prevalence). After all, *something* has to be the leading cause. Line 174-176 One way to get a multivariable sort of analysis in a study like this when there are low N problems is to use a count of symptoms. I don't know if this would help here, but it might. Tables 1 and 2 should have, in a footnote(s) what test was applied . I know it's in the text, but the tables should be self-contained. Peter Flom
--

REVIEWER	Reviewer name: Prof. Shally Awasthi Institution and Country: King George's Medical University, India Competing interests: None
REVIEW RETURNED	22-Nov-2021

GENERAL COMMENTS	This is a well conducted study. Specific comments are as follows: 1. Abstract- Conclusion to list the 4 risk factors identified.
---

	2. Respiratory distress has various parameters. Which parameter is most predictive of an adverse outcome? Is this factor the same across age categories (3). Can a consolidated respiratory distress score be made or applied to the data collected? 3. Other studies have reported gender, higher RR (above WHO cutoff), anemia as risk factor for adverse outcome. Comprehensive review has not been done in the discussion 4. What method was adopted to quantify anemia and hypoxemia?
--	---

REVIEWER	Reviewer name: Dr. Prof. S K Kabra Institution and Country: All India Institute of Medical Sciences, India Competing interests: None
REVIEW RETURNED	07-Nov-2021

GENERAL COMMENTS	In this prospective observational study, authors analysed data of children who were not eligible for enrollment in CPAP IMPACT trial. They conclude that children who died were more likely to present with respiratory distress, anemia, malaria, and a lower MUAC. Over all the manuscript is fine, with some modifications it may give clear message. 1. Abstract: Authors mention in conclusion " -----our analysis identifies potentially modifiable clinical features exist that could further stratify mortality risk among hospitalized Malawian children with severe pneumonia lacking recognized high-risk conditions. " It may be better to mention risk factors for mortality were children presenting with respiratory distress, anemia, malaria, and a lower MUAC. Some of these are modifiable. What this study adds: It can be modified as suggested for conclusion. Methods: As this was pre defined study carried out along with a RCT. Better to mention it as stand alone study rather than repeatedly mentioning about RCT. Describe details of the prospective observational study. Currently, because of repeated reference to RCT is creating mor econfusion. Results: As this was stand alone study, some details of RCT may be compared in discussion. It may be better to present results like risk factors associated with mortality in children without risk factors Discussion: Author should discuss the risk factor associated with mortality and how these can be modified.
--

REVIEWER	Reviewer name: Dr. Sunil Karande Institution and Country: Seth Gordhandas Sunderdas Medical College, India Competing interests: None
REVIEW RETURNED	10-Nov-2021

GENERAL COMMENTS	The authors have attempted to identify modifiable risk factors for mortality in young children (aged 1 to 59 months) admitted with WHO defined severe pneumonia - but who do not have associated HIV infection, severe malnutrition and / or hypoxemia. The authors have identified four modifiable factors - signs of respiratory distress, anemia, associated malaria infection and a lower mid upper arm circumference (MUAC) - which increase the risk of death in these young hospitalized children having WHO defined severe pneumonia. These are my comments / suggestions which are aimed to either improve the readability of the manuscript and/or improve its scientific importance: A) Major comments:  1) Please improve the English grammar and language of the manuscript. 2) Trim the manuscript and delete repetition of information. 3) Avoid repetition of data already displayed in Tables in the text [results section]. Specific Comments:
--

- 1) Please simplify / improve the Title of the manuscript. At present it does not represent the research work done effectively. My suggestion is "Identifying new modifiable risk factors of mortality in children aged 1-59 months admitted with WHO defined severe anemia: A single centre observational cohort study from rural Malawi".
- 2) Improve the Abstract: (i) mention the statistical data in the results and (ii) clearly mention the modifiable risk factors identified in the conclusion.
- 3) In the Introduction, clearly mention the percentage of young children with WHO defined severe anemia who need hospitalization - and do not have the known risk factors for mortality. Just mentioning "a substantial proportion" is not enough. Mentioning the known exact percentage / proportion with cross-references would enhance the importance of the present study.
- 4) Clarify what is meant by the term "HIV exposure" in the present study. In my opinion the authors should have only included documented HIV negative children in the study group. Please justify inclusion of HIV exposure [a rather vague term] children in the study group.
- 5) Delete the 10 children in whom test for malaria was not done / not available and reduce the study sample to 874. Redo the entire statistical analysis of the present study with the trimmed study sample size.
- 6) Mention the sensitivity and specificity of the rapid test utilized to detect malaria infection in the present study.
- 7) Was peripheral blood smear examination for malarial infection done? It is still the gold standard for confirming malarial infection.
- 8) Can the authors add information about the type and severity [density of parasitemia] of malarial infection in their results and analyze their results accordingly? How many of the children in both groups had *P. vivax* or *P. falciparum* or both (mixed) infection? In short, what was the severity of the malarial infection? It is possible that the patient had asymptomatic parasitemia and not active malarial infection. The authors are probably recommending that, in future, testing for malaria should be done in all young children admitted with severe pneumonia - with an aim to reduce mortality. Before jumping to such an important suggestion/recommendation a thorough detailed analysis is necessary - hence this comment/suggestion.
- 9) Clarify whether the choice of antibiotics used to treat the severe pneumonia in both groups was similar or not. This is very important and needs to be mentioned.
- 10) Please mention the names of the antibiotics used and whether these was any SOP (standard operating protocol/procedure) for choosing the antibiotic depending on the age of the child. Similarly mention the details of the anti-malarial drugs used in the present study and whether there was any SOP for choosing the anti-malarial drug.
- 11) Mention a cross-reference for the Blantyre Coma Score in the text file.
- 12) How was hypoxemia documented in the present study? This important information seems to be missing. Was it done by utilizing a pulse oximeter [mention details of instrument, sensitivity / specificity] or blood gas analysis was also done? Please clarify. In the Discussion section the authors have compared their results with previous published studies. Since the authors have not done multivariate analysis in the present study - they should clearly mention whether the risk factors identified in the other studies were by bivariate analysis or multivariate analysis. At present this important information is missing.
- 13) Why did the authors not study "degree of dehydration" as a risk factor for mortality in the present study? It is a relatively easily identifiable risk factor.
- 14) In the Discussion section the authors should give crisp and accurate information - for example line 308-309 - what was the increased odds ratio of mortality?

	15) Please simplify the language of this sentence [lines 314 to 318]. I am unable to understand what the authors are trying to imply. Why are they stating / suggesting that the sample size of the present study is inadequate? In that case their results and conclusions would have no meaning. My suggestion is to entirely delete this section or rewrite it without undermining the importance of the present study. 16) In the Conclusion section, the authors have stated that "In sum, we identified four risk factors for death (signs of respiratory distress, anemia, malaria, and lower MUAC)". My suggestion is to define these four factors clearly - for example - what is implied by signs of respiratory distress, degree of anemia (Hb level), active malarial infection [degree of parasitemia; and type of malarial infection], the cut-off value to determine lower MUAC. Unless this information is clearly stated the readers of the manuscript would not be any wiser after reading the manuscript.
--	--

REVIEWER	Reviewer name: Dr. Daniel Park Institution and Country: The George Washington University, United States Competing interests: None
REVIEW RETURNED	23-Nov-2021

GENERAL COMMENTS	The manuscript by Eckerle et al., "World Health Organization-defined severe pneumonia among hospitalized children without HIV, severe malnutrition, and hypoxemia in rural Malawi: a single center prospective observational study" provides a useful overview of clinical features that may provide predictive or prognostic value for mortality risk in certain high-burden of disease settings. There are a few questions regarding generalizability and analytic approaches taken. Minor Comments  - Abstract, methods: It may be helpful to make explicitly clear how the inclusion criteria and enrollment worked between the prospective observational study and the RCT. This is clear throughout the actual manuscript, but for readers who only see the abstract, this could be a point of confusion. - Abstract, conclusion: Missing a word in the sentence. - Methods, line 175: Was there a specific approach used to determine whether regression models could not be fit? - Results, line 192: Did the study period include the entire duration of the RCT, or just participants from the timeframe during which both studies were enrolling? - Did any of the children who died have multiple of the respiratory distress signs? If one child had all 3 signs (grunting, stridor, and apnea), it is possible that some of these signs are only incidentally implicated. - The phrase "stridor at rest" and also "stridor when calm" are used in the manuscript. I would recommend keeping this consistent. - Discussion, line 296: I found this confusing, did 50% have both severe pneumonia and malaria? What is the 3% referring to? - Table 1: A minor suggestion to organize the table may be to include headers for the type of variable (i.e. demographic, clinical history, etc.). - Was adjustment for multiple testing considered? This is not necessary in an exploratory analysis such as this one, but the conclusions and generalizations made from the data should bear this in mind since you would expect to see at least one significant result by chance (false positive) among the 20 variables evaluated. General Comments  - P-values should be re-checked. The p-values for looking at associations with stridor at rest and apnea seem small and I am
--

	unable to verify these. Some other p-values are also different by my calculations. Given the small sample size in some of the cells, the Exact tests should be used. Differences may be partially due to use of one-tailed or two-tailed tests for p-values – this should be specified in the methods. - There are potentially important learnings for pneumonia diagnosis and treatment here. However, given the limited sample size and the single-site nature of the study, I wonder if the authors generalize the results too broadly or strongly. For example, by extrapolating to 100,000 child pneumonia deaths (discussion, line 249), or recommending inclusion of some of the identified risk factors as warranted (discussion, line 331). The authors correctly point out that a larger, multi-site study would be beneficial in evaluating and confirming these findings, after which broader recommendations and generalizations should be made.
--	---

VERSION 1 – AUTHOR RESPONSE

Dear Editors:

We thank the reviewers for their overall positive and constructive critiques as well as the opportunity to revise and resubmit our manuscript. We believe we have addressed all comments in the below responses and do believe the manuscript has been substantially improved as a result.

Editor in Chief Comments to Author:

Comment 1: Title amend as suggested by reviewer 3

Author Response: Title has been changed as recommended by reviewer 3. The title now reads, “Identifying modifiable risk factors for mortality in children aged 1-59 months admitted with WHO-defined severe pneumonia: a single centre observational cohort study from rural Malawi”

Comment 2: What this study adds rewrite completely. The first two are methods and the third is too general.

Author Response: ‘What this study adds’ has been rewritten as requested.

Comment 3: You identified four risk factors for death (signs of respiratory distress, anemia, malaria, and lower MUAC). This needs to be made clearer in your paper. State this in the Abstract results and conclusion and in What this study adds.

Author Response: ‘What this study adds’ has been rewritten and the abstract results and conclusion have been edited as requested.

Comment 4: Respond in full to all the reviewers. Please note reviewer 3 has given detailed advice which should be followed.

Author Response: We have responded in full to all reviewer comments as noted and have carefully considered all of the recommendations provided by reviewer 3.

Reviewer: 1

Dr. Peter Flom, Peter Flom Consulting

Comments to the Author: I confine my remarks to statistical aspects of this paper. The general approach is fine, but I do have some issues to resolve before I can recommend publication.

Comment 1: Line 47 "Elevated" compared to what population? Rich countries? Earlier years in Malawi? Other African countries?

Author response: This has been clarified as compared to high-resource settings. The text of the abstract now reads, "Although HIV-infection, severe malnutrition, and hypoxemia are associated with high mortality in children with World Health Organization (WHO)-defined severe pneumonia in sub-Saharan Africa, many do not have these conditions and yet mortality remains elevated compared to high-resource settings."

Comment 2: Line 62-64 Why are you comparing these two groups? I don't see a reason, in which case, these lines can be dropped (the trial participants were selected to have problems). (Similarly for some sections of text. Why compare these groups?) (Not statistics but this would also let the authors drop the word "observational" in many places, leading to a smoother flow of text).

Author response: The authors have debated this comment as we do see value in contextualizing the overall study population of the RCT and the parallel observational cohort together. However, we agree that having this comparison as a main objective and in the main manuscript text is distracting and as such, we have moved this comparison to the supplemental materials where it can be accessed by interested readers, and we have revised the text throughout the manuscript and abstract to reflect this shift.

Comment 3: Line 98 When you say "leading cause" please give a rate (incidence or prevalence). After all, *something* has to be the leading cause.

Author response: Thank you for this suggestion. We have edited the introduction section of the manuscript to read as below.

"Pneumonia causes approximately 800,000 deaths outside of the neonatal period among under 5 year olds worldwide and this burden is disproportionately borne in low-income and middle-income countries (LMICs)."

Comment 4: Line 174-176 One way to get a multivariable sort of analysis in a study like this when there are low N problems is to use a count of symptoms. I don't know if this would help here, but it might.

Author response: Thank you for this comment. As recommended by the reviewer we have created a new table 2. However, rather than using clinical symptoms we thought it was a better fit for our data and approach to account for the combinations of clinical risk factors (not symptoms) identified in table 1. We have accordingly added text in the results section to refer the reader to this new table.

Comment 5: Tables 1 and 2 should have, in a footnote(s) what test was applied . I know it's in the text, but the tables should be self-contained.

Authors response: We have added footnotes throughout the table as recommended by the reviewer. Please note that because of changes above, table 1 is now supplemental table 1 and table 2 is table 1.

Reviewer: 2

Dr. Prof. S K Kabra, All India Institute of Medical Sciences

Comments to the Author

Comment 1: In this prospective observational study, authors analysed data of children who were not eligible for enrollment in CPAP IMPACT trial. They conclude that children who died were more likely to present with respiratory distress, anemia, malaria, and a lower MUAC. Over all the manuscript is fine, with some modifications it may give clear message

Authors response: We thank the reviewer for their positive feedback.

Comment 2: Abstract: Authors mention in conclusion " -----our analysis identifies potentially modifiable clinical features exist that could further stratify mortality risk among hospitalized Malawian children with severe pneumonia lacking recognized high-risk condition". " It may be better to mention risk factors for mortality were children presenting with respiratory distress, anemia, malaria, and a lower MUAC. Some of these are modifiable.

Authors response: We have modified the Abstract conclusion as suggested. The abstract conclusion now reads as follows.

“In this single-center study, our analysis identifies potentially modifiable risk factors for mortality among hospitalized Malawian children with severe pneumonia: specific signs of respiratory distress (grunting, stridor, apnea), hemoglobin <5 g/dL, and malaria infection. Significant differences in mean hemoglobin and MUAC were observed between those who survived and those who died. These factors could further stratify mortality risk among hospitalized Malawian children with severe pneumonia lacking recognized high-risk conditions.”

Comment 3: What this study adds: It can be modified as suggested for conclusion.

Authors response: We have modified the “What this study adds” as suggested. The “What this study adds” now reads as follows.

- “About 20% of pneumonia deaths at one rural district hospital in Malawi over two years were attributable to children lacking HIV, severe malnutrition, and hypoxemia.

- Among Malawian children with severe pneumonia without HIV, severe malnutrition, and hypoxemia, respiratory distress (apnea, stridor, grunting), hemoglobin <5 g/dL, and malaria predicted mortality.
- Significant differences in mean hemoglobin and mid-upper arm circumference were present in severe pneumonia cases lacking recognized high-risk conditions who survived or died.”

Comment 4: Methods: As this was pre-defined study carried out along with a RCT, better to mention it as standalone study rather than repeatedly mentioning about RCT.

Authors response: Thank you for this suggestion. As noted in a prior response we have move the RCT comparisons to the supplemental material for interested readers and revised the text throughout to reflect this change.

Comment 5: Describe details of the prospective observational study. Currently, because of repeated reference to RCT is creating more confusion.

Authors response: We have expanded the details of the prospective observational study throughout the methods section.

Comment 6: Results: As this was standalone study, some details of RCT may be compared in discussion. It may be better to present results like risk factors associated with mortality in children without risk factors

Authors response: As previously noted the references to the RCT have been noted only to provide context to the observational study and have otherwise been generally confined to discussion only. The remainder of text has been modified to improve clarity.

Comment 7: Discussion: Author should discuss the risk factor associated with mortality and how these can be modified.

Authors response: The current text in the discussion includes the following language, which discusses how children identified to have any of these risk factors may could be more optimally managed and their outcomes modified.

“It is possible these children, if appropriately identified, may be more responsive to treatments and more likely to survive; the ability to target care would be beneficial. For example, such children may be more successfully managed in hospital units with higher nurse to patient ratios that allow closer monitoring and more meticulous care. Results from CPAP IMPACT suggest meaningful reductions in mortality among higher risk children with severe pneumonia may be challenging to achieve with available approaches in LMICs.”

We have also added additional text to the discussion conclusion to further address this comment from the reviewer.

“Based on these findings systematic screening of children with severe pneumonia for additional signs of respiratory distress, checking of hemoglobin levels, testing for malaria (when endemic), and measuring MUAC, followed by closer monitoring and standard treatment for any identified abnormalities, seem feasible and warrant further consideration.”

Reviewer: 3

Dr. Sunil Karande, Seth Gordhandas Sunderdas Medical College

Comments to the Author: The authors have attempted to identify modifiable risk factors for mortality in young children (aged 1 to 59 months) admitted with WHO defined severe pneumonia - but who do not have associated HIV infection, severe malnutrition and / or hypoxemia. The authors have identified four modifiable factors - signs of respiratory distress, anemia, associated malaria infection and a lower mid upper arm circumference (MUAC) - which increase the risk of death in these young hospitalized children having WHO defined severe pneumonia.

These are my comments / suggestions which are aimed to either improve the readability of the manuscript and/or improve its scientific importance:

A) Major comments:

Comment 1: Please improve the English grammar and language of the manuscript.

Authors response: We have reviewed the writing of the manuscript in detail overall and have revised for clarity.

Comment 2: Trim the manuscript and delete repetition of information.

Authors response: We have reviewed the writing of the manuscript and have removed any areas of unnecessary repetition. Please note that it is both our writing style and belief that some repetition of key points is good scientific writing, and thus we have maintained some repetition throughout the manuscript where we deemed appropriate to stress key points for the reader.

Comment 3: Avoid repetition of data already displayed in Tables in the text [results section].

Authors response: We have substantially edited the results section and removed the majority of the text related to the prior Table 1 (now a supplemental table).

Specific Comments:

Comment 4: Please simplify / improve the Title of the manuscript. At present it does not represent the research work done effectively. My suggestion is "Identifying new modifiable risk factors of mortality in children aged 1-59 months admitted with WHO defined severe anemia: A single centre observational cohort study from rural Malawi".

Authors response: We assume that the reviewer meant severe pneumonia rather than severe anemia. We have otherwise changed the title as recommended by the reviewer.

Comment 5: Improve the Abstract: (i) mention the statistical data in the results and (ii) clearly mention the modifiable risk factors identified in the conclusion.

Authors response: The journal style specifically requests authors to not include p values in the results section of the abstract (see https://bmjpaedsopen.bmj.com/pages/authors/#original_research : “the most important results illustrated by numerical data but not p values”). As a result we have kept the language as written based on this direction but we can add p values to the results section of the abstract should this be desired by the editor. We have stated the potentially modifiable risk factors in the abstract conclusion as requested.

Comment 6: In the Introduction, clearly mention the percentage of young children with WHO defined severe anemia who need hospitalization - and do not have the known risk factors for mortality. Just mentioning "a substantial proportion" is not enough. Mentioning the known exact percentage / proportion with cross-references would enhance the importance of the present study.

Authors response: We assume that the reviewer meant severe pneumonia rather than severe anemia. As such, we have revised the introduction for clarity to add specific data as recommended by the reviewer. Importantly, we note that there is uncertainty in the epidemiological data for children lacking already recognized risk factors, which is one of the justifications for the research. Our edits reflect this context. The relevant text of the introduction now reads as follows:

“While children with severe pneumonia but without these conditions may have relatively lower mortality, they still may comprise a substantial overall burden of pneumonia morbidity and mortality in LMICs.¹ For example, in Malawi approximately half of child pneumonia deaths were attributable to HIV and almost 90% of hospitalized pneumonia were among children without hypoxemia.^{4,5} In a multi-country severe pneumonia study 61% of deaths occurred among children without severe malnutrition.⁶”

We have added the following 3 citations to support the above statement.

Theodoratou E, McAllister DA, Reed C, et al. Global, regional, and national estimates of pneumonia burden in HIV-infected children in 2010: a meta-analysis and modelling study. *Lancet Infect Dis* 2014; **14**: 1250–58.

McCollum ED, Nambiar B, Deula R, et al. Impact of the 13-valent pneumococcal conjugate vaccine on clinical and hypoxemic childhood pneumonia over three years in central Malawi: an observational study. *PLoS One* 2017; **12**: e0168209.

Gallagher KE, Knoll MD, Prospero C, et al. The predictive performance of a pneumonia severity score in HIV-negative children presenting to hospital in seven low and middle-income countries. *Clin Infect Dis* 2020; **70**: 1050–1057.

Comment 7: Clarify what is meant by the term "HIV exposure" in the present study. In my opinion the authors should have only included documented HIV negative children in the study group. Please justify inclusion of HIV exposure [a rather vague term] children in the study group.

Authors response: Please note that HIV exposed or HIV infected children were not included in the observational cohort. Children included in the observational cohort were all HIV negative (uninfected and unexposed). HIV exposure was defined in accordance with Malawi national guidelines at the time of the study, which defined HIV exposure as a <24 month old child born to a HIV-infected mother and not meeting HIV-infection criteria. HIV infection is defined as a <12 month old with a positive HIV DNA PCR or ≥ 12 month old with HIV antibodies. We have added these definitions to the methods section of the manuscript for clarity. The relevant section reads as follows:

“Children were eligible if 1-59 months with WHO-defined severe pneumonia and without any of three high-risk conditions: (1) HIV-infection (<12 month old with a positive HIV DNA PCR or ≥ 12 month old with HIV antibodies) or exposure (<24 month old child born to a HIV-infected mother and not meeting HIV-infection criteria), (2) severe malnutrition (weight for age z score < -3 standard deviations (SD), mid-upper arm circumference (MUAC) <11.5 cm, and/or bilateral pedal edema), or (3) a SpO₂ <90%.”

Comment 8: Delete the 10 children in whom test for malaria was not done / not available and reduce the study sample to 874. Redo the entire statistical analysis of the present study with the trimmed study sample size.

Authors response: Thank you for this suggestion. We have carefully considered this comment from the reviewer. However, rather than deleting children from our primary analysis we have provided a new sensitivity analysis as a supplemental table with patients removed for whom no malaria test was available, as requested by the reviewer. We did observe that in this sensitivity analysis that the quantitative association between apnea and death changed (the relationship did not reach statistical significance) but we did not observe any qualitative change in the relationship. None of the other findings observed in the primary analysis changed. We have added this finding to the results section, which reads as follows:

“In a sensitivity analysis excluding seven children without a malaria test result, the overall findings were consistent except for the relationship between apnea and death. This association was qualitatively similar but did not reach statistical significance (p=0.16, Supplemental Table 1).”

Comment 9: Mention the sensitivity and specificity of the rapid test utilized to detect malaria infection in the present study.

Authors response: The test characteristics relative to *p. falciparum* were added. The relevant text in the methods section now reads as follows.

“After enrollment, participants were assessed for malaria (SD Bioline (HRP2/pLDH, sensitivity 97.4-99.7%, specificity 99.3-99.7%) rapid antigen tests (Abbott, Johannesburg, South Africa)) and anemia (Hemocue 301+).”

Comment 10: Was peripheral blood smear examination for malarial infection done? It is still the gold standard for confirming malarial infection.

Authors response: We agree that peripheral blood smear microscopy is the gold standard for malaria diagnosis. However, children in this study were by design treated according to hospital protocols and guidelines in Malawi, which provided only for routine rapid diagnostic tests at admission; blood smear results, if obtained, were not recorded. It is important to further stress that smear microscopy capacity is limited in Malawi, and this (along with local guidelines) informed our approach. We have added this information into the methods section and also as a limitation to the discussion section. The relevant text now reads as below:

Methods –

“Consistent with national guidelines malaria infection was confirmed on all participants at enrollment using rapid antigen testing (SD Bioline (HRP2/pLDH, sensitivity 97.4%-99.7%, specificity 99.3%-99.7%)) as routine microscopy was limited.¹³”

Discussion –

“First, malaria detection was determined by rapid antigen testing rather than peripheral blood microscopy. While children in endemic settings may have a positive rapid test due to persistent antigenemia rather than an acute infection,²⁷ our use of rapid testing was consistent with local practice in Malawi, a setting with limited microscopy capacity.¹³”

We added the below citation (citation number 27):

Mayxay M, Pukrittayakamee S, Chotivanich K, Looareesuwan S, White NJ. Persistence of Plasmodium falciparum HRP-2 in successfully treated acute falciparum malaria. *Trans R Soc Trop Med Hyg* 2001;**95**:179-82.

Comment 11: Can the authors add information about the type and severity [density of parasitemia] of malarial infection in their results and analyze their results accordingly? How many of the children in both groups had *P. vivax* or *P. falciparum* or both (mixed) infection? In short, what was the severity of the malarial infection? It is possible that the patient had asymptomatic parasitemia and not active malarial infection. The authors are probably recommending that, in future, testing for malaria should be done in all young children admitted with severe pneumonia - with an aim to reduce mortality. Before jumping to such an important suggestion/recommendation a thorough detailed analysis is necessary - hence this comment/suggestion.

Authors Response: Please see the above response as well as the previously detailed additions to the methods and discussion sections. We agree that accurate determination of malaria infection is best done with peripheral blood smear microscopy and that it is probable that some unknown proportion of children

in this study could have persistent antigenemia rather than acute malaria infection. Because of this, we have not made definitive recommendations about routine malaria testing for all children with pneumonia. It is also possible that there is a collinear relationship between malaria and anemia that cannot be fully explored in this data. But given the prevalence of malaria in the study setting, and the possibility for symptomatic malaria infection to be incorrectly interpreted as severe pneumonia because of respiratory distress, the authors believe it is important to highlight the study finding, even in the absence of blood smear results. We believe that our overall conclusion is balanced and appropriately caveated. Please see the text below from the conclusion of the discussion section.

“Based on these findings systematic screening of children with severe pneumonia for additional signs of respiratory distress, checking of hemoglobin levels, testing for malaria (when endemic), and measuring MUAC, followed by closer monitoring and standard treatment for any identified abnormalities, seem feasible and warrant further consideration. Intervention studies examining the modifiability of these risk factors in similar populations are needed as is larger, confirmatory observational research in other LMIC settings.”

Comment 12: Clarify whether the choice of antibiotics used to treat the severe pneumonia in both groups was similar or not. This is very important and needs to be mentioned.

Authors response: As we have indicated in the methodology section of the manuscript standard care protocols were followed for all patients participating in the observational study. It was beyond the scope and capacity of the study to record specific treatments after enrollment. Given pneumonia care is highly protocolized in this setting the authors have no reason to suspect that this was not the case for this population. We have expanded the methodology section to detail this point. The text reads as below.

“For observational participants study staff recommended initial evaluations and treatments but thereafter children received usual care per local protocols administered by government hospital staff, including oxygen, antibiotics, and fluids. Per local protocols, patients received benzylpenicillin and gentamicin for initial pneumonia treatment, and were switched to ceftriaxone for treatment failure. Similarly, per local protocols, severe malaria was treated with artesunate and uncomplicated malaria (as determined by the treating clinician) was treated with lumefantrine-artemether.”

Comment 13: Please mention the names of the antibiotics used and whether there was any SOP (standard operating protocol/procedure) for choosing the antibiotic depending on the age of the child. Similarly mention the details of the anti-malarial drugs used in the present study and whether there was any SOP for choosing the anti-malarial drug.

Authors response: Please see our response to the prior comment and the additional information we have added to the methodology section.

Comment 14: Mention a cross-reference for the Blantyre Coma Score in the text file.

Authors response: We have added the below reference to the manuscript, which is a commonly used clinical reference in Malawi.

A PAEDIATRIC HANDBOOK for Malawi. Third Edition. 2008. Available at:
http://cms.kcn.unima.mw:8002/moodle/file.php/1/Phillips_Kazembe_Paediatric_Handbook_for_Malawi_2008.pdf

Comment 15: How was hypoxemia documented in the present study? This important information seems to be missing. Was it done by utilizing a pulse oximeter [mention details of instrument, sensitivity / specificity] or blood gas analysis was also done? Please clarify.

Authors response: The peripheral arterial oxyhemoglobin saturation was measured using a Masimo Rad 5 pulse oximeter. This has been added into the methodology section. Test characteristics of the device on pediatric patients are not readily available but the device has met all appropriate performance and certification criteria. The methodology text reads as below:

“SpO₂ was measured while the child was breathing in room air using a Rad5 pulse oximeter (Masimo, Irvine, CA, USA).”

Comment 16: Why did the authors not study "degree of dehydration" as a risk factor for mortality in the present study? It is a relatively easily identifiable risk factor.

Authors response: Thank you for this comment. In general, our intention was to evaluate more objective clinical measurements not currently emphasized in the WHO guidelines as risk factors for mortality. As such, tachycardia is more objective assessment for dehydration than other existing subjective approaches like assessing skin turgor, observing for sunken eyes, capillary refill, etc. We have better emphasized this general focus throughout this revised version of the manuscript.

Comment 17: In the Discussion section the authors should give crisp and accurate information - for example line 308-309 - what was the increased odds ratio of mortality?

Authors response: Thank you for this comment. We have added statistics to the sentence as suggested by the reviewer. The sentence now reads as below.

“Unlike this study, prior inpatient pediatric studies from Malawi found associations between moderately low SpO₂ ranges and increased odds of mortality among children with pneumonia (between 50% and 330%).”

Comment 18: Please simplify the language of this sentence [lines 314 to 318]. I am unable to understand what the authors are trying to imply. Why are they stating / suggesting that the sample size of the present study is inadequate? In that case their results and conclusions would have no meaning. My suggestion is to entirely

delete this section or rewrite it without undermining the importance of the present study.

Authors response: We have revised this paragraph for clarity. It now reads as follows:

“Although in our study we also observed higher point estimates for mortality among children with a SpO₂ 90-92%, compared to 93-100%, this association did not reach statistical significance. We would expect that with a larger sample size a similar association between SpO₂ and mortality would have been observed in this study.”

Comment 19: In the Conclusion section, the authors have stated that "In sum, we identified four risk factors for death (signs of respiratory distress, anemia, malaria, and lower MUAC)". My suggestion is to define these four factors clearly - for example - what is implied by signs of respiratory distress, degree of anemia (Hb level), active malarial infection [degree of parasitemia; and type of malarial infection], the cut-off value to determine lower MUAC. Unless this information is clearly stated the readers of the manuscript would not be any wiser after reading the manuscript.

Authors response: We have revised the discussion section to further stress the potential importance of cohorting these patients to improve monitoring and care.

“For example, such children may be more successfully managed by cohorting them into hospital units with higher nurse to patient ratios that allow closer monitoring and more meticulous care.”

We have also further edited the conclusion section for clarity and have added some of these specific points where appropriate. We feel that it is important to stress that this work sets the foundation for additional research and, while an important addition to the literature, it should not be considered definitive given it is an observational design. This is why we have recommended intervention studies designed to assess the modifiability of these identified risk factors as well as larger studies in varying populations. Of note, because specific cutpoints for MUAC were not identified (there was a significant difference between mean MUAC among those who survived and those who died, but our analyses were unable to establish clinically useful lower limits), this conclusion was modified. The conclusion section now reads as below.

“In sum, we identified multiple risk factors for death (specific signs of respiratory distress (stridor, apnea, grunting), hemoglobin <5 g/dL, and malaria) among Malawian children with severe pneumonia but without severe malnutrition, HIV infection or exposure, and hypoxemia. Children who died had a lower MUAC than those who survived, but statistically significant thresholds were not apparent. Based on these findings systematic screening of children with severe pneumonia for additional signs of respiratory distress, checking of hemoglobin levels, testing for malaria (when endemic), and measuring MUAC, followed by closer monitoring and standard treatment for any identified abnormalities, seem feasible and warrant further consideration. Intervention studies examining the modifiability of these risk factors in similar populations are needed as is larger, confirmatory observational research in other LMIC settings.”

Reviewer: 4
Prof. Shally Awasthi, King George's Medical University

Comments to the Author

This is a well conducted study.

Authors response: Thank you for this positive feedback.

Specific comments are as follows:

Comment 1: Abstract- Conclusion to list the 4 risk factors identified.

Authors response: As detailed in prior responses we have revised the conclusion to the abstract to list the risk factors.

Comment 2: Respiratory distress has various parameters. Which parameter is most predictive of an adverse outcome? Is this factor the same across age categories (3). Can a consolidated respiratory distress score be made or applied to the data collected?

Authors response: We found that apnea, stridor and grunting were predictive of death among our cohort. We attempted to analyze these 3 factors to determine whether it was a single factor versus a combination which predicted death, but very few children had more than one factor (7 of 883). Therefore, we were unable to draw broader conclusions or create a score.

Comment 3: Other studies have reported gender, higher RR (above WHO cutoff), anemia as risk factor for adverse outcome. Comprehensive review has not been done in the discussion

Authors response: We have added references and expanded relevant areas of the discussion as recommended by the reviewer.

Comment 4: What method was adopted to quantify anemia and hypoxemia?

Authors response: As detailed in prior responses to reviewers we have added these points to the methodology section.

Reviewer: 5
Dr. Daniel Park, The George Washington University

Comments to the Author

The manuscript by Eckerle et al., "World Health Organization-defined severe pneumonia among hospitalized children without HIV, severe malnutrition, and hypoxemia in rural Malawi: a single center prospective observational study" provides a useful overview of clinical features that may provide predictive or prognostic value for mortality risk in certain high-burden of disease settings. There are a few questions regarding generalizability and analytic approaches taken.

Authors response: Thank you for this overall positive feedback.

Minor Comments

Comment 1: Abstract, methods: It may be helpful to make explicitly clear how the inclusion criteria and enrollment worked between the prospective observational study and the RCT. This is clear throughout the actual manuscript, but for readers who only see the abstract, this could be a point of confusion.

Authors response: Based on the comments of other reviewers, references to the RCT have largely been removed from the abstract other than for context of this study.

Comment 2: Abstract, conclusion: Missing a word in the sentence.

Authors response: Thank you, this section has been edited.

Comment 3: Methods, line 175: Was there a specific approach used to determine whether regression models could not be fit?

Authors response: We made this decision based on the widely used statistical 'one in ten rule' that maintains that one predictive variable can be used for every 10 events of interest. Since our dataset had 21 deaths this restricted any model to 2 predictor variables, which we felt was insufficient to move forward with a reasonable multivariable model.

Comment 4: Results, line 192: Did the study period include the entire duration of the RCT, or just participants from the timeframe during which both studies were enrolling?

Authors response: The results reported are from the timeframe during which both studies were enrolling. There was a period of time where the RCT was enrolling but the observational study had not yet commenced. Both studies ended at the same time.

Comment 5: Did any of the children who died have multiple of the respiratory distress signs? If one child had all 3 signs (grunting, stridor, and apnea), it is possible that some of these signs are only incidentally implicated.

Authors response: Thank you for raising this point. Other reviewers have provided the same comment and we would direct you to those detailed responses. In short, only 7 of 883 children had more than one of these three respiratory signs, which limited our ability to draw any broader conclusions about combinations of respiratory symptoms.

Comment 6: The phrase "stridor at rest" and also "stridor when calm" are used in the manuscript. I would recommend keeping this consistent.

Authors response: The manuscript has been edited for consistency.

Comment 7: Discussion, line 296: I found this confusing, did 50% have both severe pneumonia and malaria? What is the 3% referring to?

Authors response: Thank you for identifying this error. The discussion of this reference has been edited. This sentence now reads as below.

“A large Kenyan cohort of hospitalized hypoxemic children found a final diagnosis of malaria more frequently than lower respiratory tract infection (35% versus 32%).²²”

Comment 8: Table 1: A minor suggestion to organize the table may be to include headers for the type of variable (i.e. demographic, clinical history, etc.).

Authors response: Thank you for this suggestion, this Table has been modified to reflect this helpful suggestion.

Comment 9: Was adjustment for multiple testing considered? This is not necessary in an exploratory analysis such as this one, but the conclusions and generalizations made from the data should bear this in mind since you would expect to see at least one significant result by chance (false positive) among the 20 variables evaluated.

Authors response: Multiple testing was not considered because of relatively small mortality events and low power, to avoid the possibility of type II error.

General Comments

Comment 10: P-values should be re-checked. The p-values for looking at associations with stridor at rest and apnea seem small and I am unable to verify these. Some other p-values are also different by my calculations. Given the small sample size in some of the cells, the Exact tests should be used. Differences may be partially due to use of one-tailed or two-tailed tests for p-values – this should be specified in the methods.

Authors response: Analysis was redone using both STATA and R to compare p-values. Exact tests used have been listed and also footnoted in the table itself. This information is included at the end of this response (please see below).

Comment 11: There are potentially important learnings for pneumonia diagnosis and treatment here. However, given the limited sample size and the single-site nature of the study, I wonder if the authors generalize the results too broadly or strongly. For example, by extrapolating to 100,000 child pneumonia deaths (discussion, line 249), or recommending inclusion of some of the identified risk factors as warranted (discussion, line 331). The authors correctly point out that a larger, multi-site study would be beneficial in evaluating and confirming these findings, after which broader recommendations and generalizations should be made.

Authors response: Thank you for this comment. We have revised the discussion section to further soften the generalization of the findings. See below for the revised sentences in the discussion section.

“Although further research is needed in other settings, extrapolating these observations to broader estimates suggests that potentially high numbers of child pneumonia deaths annually in sub-Saharan Africa may occur among children without severe malnutrition, HIV, and hypoxemia.”

“Altogether, our results imply children with severe pneumonia lacking HIV, severe malnutrition, or hypoxemia may be important in the broader child pneumonia epidemiological context and could benefit from more systematic scrutiny of four potentially modifiable domains of risk, (1) respiratory distress, (2) anemia, (3) malaria, and, (4) non-severe malnutrition, through more rigorous screening and testing.”

“Altogether, given the co-incidence of these conditions, the mortality risk associated with severe malaria and anemia, and the limited performance of pallor, routine hemoglobin and malaria testing among children with difficulty breathing merits further evaluation.”

“Based on these findings systematic screening of children with severe pneumonia for additional signs of respiratory distress, checking of hemoglobin levels, testing for malaria (when endemic), and measuring MUAC, followed by closer monitoring and standard treatment for any identified abnormalities, seem feasible and warrants further assessment. Intervention studies examining the modifiability of these risk factors in similar populations are needed as is larger, confirmatory observational research in other settings.”

Stata and R output for Reviewer 5 (Dan Park), Comment 10:

Demonstration of the Fisher Exact tests in two statistical programs, Stata (used for this paper) and R (to confirm some of the most significant p-values).

Example 1: Comparing death (variable name obj3survived0died1) by malaria positivity (variable name: ml_res). Fisher Exact p-values highlighted in each program's output.

Stata output:

+-----+

```
| Key |
|-----|
| frequency |
| column percentage |
|-----|
```

ml_res	obj3survived0died1		Total
	0	1	
1	663	10	673
	77.45	47.62	76.74
2	193	11	204
	22.55	52.38	23.26
Total	856	21	877
	100.00	100.00	100.00

```
Fisher's exact = 0.003
1-sided Fisher's exact = 0.003
```

```
. di r(p_exact)
.00316854
```

R output:

```
> data<-matrix(c(663,193,10,11), nrow=2)
> data
      [,1] [,2]
[1,] 663  10
[2,] 193  11
> fisher.test(data)
```

Fisher's Exact Test for Count Data

```
data: data
p-value = 0.003169
```

Example 2: Comparing death (variable name obj3survived0died1) by grunt (variable name: grunt). Fisher Exact p-values highlighted in each program's output.

Stata output:

```
+-----+
| Key |
|-----|
| frequency |
| column percentage |
|-----|
```

grunt	obj3survived0died1		Total
	0	1	
0	754	11	765
	88.19	52.38	87.33

1	101	10	111
	11.81	47.62	12.67
Total	855	21	876
	100.00	100.00	100.00

Fisher's exact = 0.000
1-sided Fisher's exact = 0.000

```
. di r(p_exact)
.00007608
```

```
R output:
> data<-matrix(c(754,101,11,10), nrow=2)
> data
      [,1] [,2]
[1,] 754  11
[2,] 101  10
> fisher.test(data)
```

Fisher's Exact Test for Count Data

```
data: data
p-value = 7.608e-05
```

VERSION 2 – REVIEW

REVIEWER	Reviewer name: Dr. Peter Flom Institution and Country: Peter Flom Consulting, United States Competing interests: None
REVIEW RETURNED	06-Jan-2022

GENERAL COMMENTS	The authors have addressed my concerns and I now recommend publication.
---

REVIEWER	Reviewer name: Dr. Sunil Karande Institution and Country: Seth Gordhandas Sunderdas Medical College, India Competing interests: None
REVIEW RETURNED	11-Jan-2022

GENERAL COMMENTS	The authors have replied satisfactorily to my comments. I regret the confusion caused by my writing 'anemia' instead of 'pneumonia' in a few comments.
--

REVIEWER	Reviewer name: Dr. Daniel Park Institution and Country: The George Washington University, United States Competing interests: None
REVIEW RETURNED	25-Jan-2022

GENERAL COMMENTS	The authors have done a commendable job responding to all suggestions and comments. I have no additional comments on the manuscript itself, but have one remaining point of clarification. Comments: - With regards to confirmation of the p-values, the variables noted on the last review were stridor at rest and apnea. These have two-
---

	tailed Fisher Exact Test p-values of 0.08 and 0.31 (or Mid-P exact test p-values of 0.04 and 0.16) based on the numbers provided in Table 1 (which are both reported as $p < 0.05$). Similarly, Hb < 5 g/dL have Exact p-values of 0.047 (Fisher) and 0.03 (Mid-P), which are different from the p-values in Table 1. I also noted that numbers in the Stata output (in the response to reviewers) are slightly different from the values provided in Table 1. For example, among survivors, the Stata output shows 101 with grunting and 754 without, whereas in the Table it shows 102 with grunting and 760 without. Perhaps that may be the reason for the different results?
--	--

VERSION 2 – AUTHOR RESPONSE

We thank the editors and reviewers for their ongoing consideration of our manuscript and to Dr. Park for his detailed review. We apologize for the confusion regarding this point and have attempted to respond point-by-point to clarify. For context, Dr. Park’s relevant comment from the initial review is included below:

Reviewer 3: Dr. Daniel Park, The George Washington University

Comment (from initial review): P-values should be re-checked. The p-values for looking at associations with stridor at rest and apnea seem small and I am unable to verify these. Some other p-values are also different by my calculations. Given the small sample size in some of the cells, the Exact tests should be used. Differences may be partially due to use of one-tailed or two-tailed tests for p-values – this should be specified in the methods.

Comment (from current review): The authors have done a commendable job responding to all suggestions and comments. I have no additional comments on the manuscript itself but have one remaining point of clarification. With regards to confirmation of the p-values, the variables noted on the last review were stridor at rest and apnea. These have two-tailed Fisher Exact Test p-values of 0.08 and 0.31 (or Mid-P exact test p-values of 0.04 and 0.16) based on the numbers provided in Table 1 (which are both reported as $p < 0.05$).

Authors response: In version 1 of the manuscript submission, differences in characteristics between those who survived and died were listed in Table 2 (this was renamed ‘Table 1’ in the subsequent version 2 of the manuscript). In version 1, the p-values for stridor at rest and apnea were, respectively, < 0.01 and 0.04. Chi-square testing was used to obtain these values.

Based on the reviewer comments, the data presented in the current table 1 has now been re-analyzed using Wilcoxon rank sum, Fisher’s exact, or two sample t-tests where appropriate (each of which is denoted with superscripts). There were several differences in p-values, all of which have been edited with tracked changes. Of note, the variable ‘apnea’ is no longer statistically significant using Fisher’s exact test and this has been accordingly edited throughout the manuscript and Table 2 has been reanalyzed without apnea as a risk factor.

Comment from reviewer: Similarly, Hb < 5 g/dL have Exact p-values of 0.047 (Fisher) and 0.03 (Mid-P), which are different from the p-values in Table 1. I also noted that numbers in the Stata output (in the response to reviewers) are slightly different from the values provided in Table 1.

Authors response:

* REGARDING Hb < 5 variable

We thank the reviewer for noting this. The reviewer comments on the tabulated numbers in Table 1 being different from the Stata output for this variable. For convenience we repeat the Stata output for this variable below:

For Main Table 1:

| col

Hgb <5 | survived died | Total

```

-----+-----+-----
yes | 24 3 | 27
no | 839 18 | 857
-----+-----+-----
Total | 863 21 | 884

```

Fisher's exact = 0.023
1-sided Fisher's exact = 0.023

We have already discussed the issue of nearly equal one-sided and two-sided p-values at the edge. In the Word document for Table1, our prior version had $p < 0.01$. We will now correct it to 0.02

Reviewer comment: For example, among survivors, the Stata output shows 101 with grunting and 754 without, whereas in the Table it shows 102 with grunting and 760 without. Perhaps that may be the reason for the different results?

Authors response: We apologize for the confusion regarding Stata output presented to the reviewer: as noted above, in the last version of response to reviewer, we had demonstrated the Stata output for grunting related to 'Supplementary Table 1', which is a subset of the data.

The reviewer correctly noticed that this differs from the numbers tabulated in Main Table 1. We thus re-demonstrate the Stata output for Main Table 1:

```

| col
grunting | survived died | Total
-----+-----+-----
yes | 102 10 | 112
no | 760 11 | 771
-----+-----+-----
Total | 862 21 | 883

```

Fisher's exact = 0.000
1-sided Fisher's exact = 0.000

```

. * Two-sided p-value
. di r(p_exact)
.00007694
. * One-sided p-value
. di r(p1_exact)
.00007694

```

NOTE: The two sided p-value is < 0.01 as correctly tabulated in the Word document of the manuscript. Of note, relative the total $N=884$ reported for subjects in Table 1, the specific variable of 'grunting' was missing information for one participant ($N=883$). This has been added in a footnote in Table 1.

VERSION 3 – REVIEW

REVIEWER	Reviewer name: Dr. Peter Flom Institution and Country: Peter Flom Consulting, United States Competing interests: None
REVIEW RETURNED	21-Mar-2022
GENERAL COMMENTS	The authors have responded to my concerns and I now recommend publication. Peter Flom

REVIEWER	Reviewer name: Dr. Sunil Karande Institution and Country: Seth Gordhandas Sunderdas Medical College, India Competing interests: None
REVIEW RETURNED	18-Mar-2022

GENERAL COMMENTS	The Authors have addressed my comments satisfactorily. I appreciate that the Authors have stated that the present research study will set the foundation for additional research and that it should not be considered definitive given that it is an observational study.
---

REVIEWER	Reviewer name: Dr. Daniel Park Institution and Country: The George Washington University, United States Competing interests: None
REVIEW RETURNED	25-Mar-2022

GENERAL COMMENTS	The authors have thoroughly responded to all suggestions and have provided excellent clarifications. All applicable changes have been made to the manuscript, which reads well. I have no further comments and recommend that the paper be accepted.
--

VERSION 3 – AUTHOR RESPONSE